# Versatile Integrated Polarizers Based on Geometric Metasurfaces

**DOI:** 10.3390/nano12162816

**Published:** 2022-08-17

**Authors:** Zhiyuan Yue, Jilian Xu, Peiyao Lu, Shuyun Teng

**Affiliations:** Shandong Provincial Engineering and Technical Center of Light Manipulations & Shandong Provincial Key Laboratory of Optics and Photonic Device, School of Physics and Electronics, Shandong Normal University, Jinan 250014, China

**Keywords:** metasurface, polarization, vector beam

## Abstract

We propose versatile integrated polarizers based on geometric metasurfaces. Metasurface polarizer consists of an L-shaped hole array etched on a silver film, and it can simultaneously generate several polarization states, including linear polarization, circular polarization, elliptical polarization, or even hybrid polarization. Meanwhile, the combination of output polarization states changes with the illumination polarization type. The theoretical analysis provides a detailed explanation for the generation of the integrated polarization states. The well-designed metasurface polarizers may generate more complex polarization modes, including vector beams and vector vortex beams. The theoretical and simulated results confirm the polarization performance of the proposed integrated metasurface polarizers. The compact design of metasurface polarizers and the controllable generation of versatile polarization combinations are a benefit to the applications of polarization light in optical imaging, biomedical sensing, and material processing.

## 1. Introduction

Polarization is the most basic property of a light beam. The differently polarized light has different applications, and flexible manipulation of the polarization state of light is desired for practical applications. Uniformly polarized light beams, such as linearly polarized and elliptically polarized and circularly polarized beams, are the common polarized ones, and they are usually used in optical imaging and information processing. In recent years, the vector beams with spatially varying polarization states have attracted much attention, and many exotic phenomena, such as the production of a strong longitudinal field component under tightly focused conditions [1,2] and the smaller focal spot [3,4], have been explored. The tight focusing makes the vector beam show great advantages in optical trapping [5,6], high-resolution imaging [7,8], particle acceleration [9], optical microscopy [10,11,12,13], and other fields. Undoubtedly, the multifunctional polarizer that outputs simultaneously multiple polarization states is fascinating for practical applications.

Uniformly polarized light beams can be generated by diffraction elements and refraction elements. The main methods to generate vector beams are subwavelength grating [14,15], orientation-tailored liquid crystal [16,17,18], interferometer [19,20], laser in-cavity device [21], and fiber laser [22,23]. These elements are obviously difficult to integrate. Metasurface consisting of nanounits offers great potential for generating polarized beams, which is an efficient way to manipulate vector beams [24,25]. Due to the ultra-thin thickness, simple production, easy integration, and light manipulation on a nanometer scale, metasurfaces have been widely used in the design of optical elements, such as circular polarization analyzers [26,27], polarization converters [28,29], optical vortices [30,31]. Recently, our group has proposed a compact metasurface structure to realize the polarization transformation from a uniform polarization state to a hybrid polarization state [32]. These works indicate that metasurface can effectively manipulate the polarization of light. However, a metasurface integrated polarizer has been little studied, and the changeable multiple polarization output produced by a single metasurface structure is desired in practical applications.

In this paper, we propose versatile integrated polarizers based on optical metasurfaces. It can produce a variety of polarization states, including linear polarization, circular polarization, elliptical polarization, and even mixed polarization. The combination of output polarization states varies with the illumination polarization type. Theoretical analysis provides the basis for the generation of multiple polarization states. Through optimizing the structure parameters of L-shaped nanohole, the equivalent quarter-wave plate is obtained. Then the versatile integrated polarizer is composed of optimized L-shaped nanoholes, and the output polarization combination of metasurface confirms the performance of the versatile integrated polarizer. Except for the combination of uniform polarization states, the design principle of this paper is also available to generate the nonuniform polarization states, including vector beams and vector vortex beams, using the will-designed metasurface polarizers. The compact design of the metasurface polarizer and the generation of the integrated polarization states are beneficial to the applications of polarized light in optical imaging, biomedical sensing, and material processing. 

## 2. Design Principle

As we know, one quarter-wave plate can be expressed by the Jones matrix
(1)$$T=\left(\begin{array}{cc} \cos^{2}\alpha +i\;\sin^{2}\alpha & (1-i)\sin\alpha \cos\alpha \\ (1-i)\sin\alpha \cos\alpha & \sin^{2}\alpha +i\;\cos^{2}\alpha \end{array}\right)$$
with *α* denoting the cross angle of its fast axis and *x* axis. Under the illumination by one linearly polarized beam with the polarization angle of *γ*, the transmission field can be obtained,
(2)$$E(\alpha ,\gamma )=\left(\begin{array}{c} \cos\alpha \cos(\gamma -\alpha )-i\sin\alpha \sin(\gamma -\alpha )\\ \sin\alpha \cos(\gamma -\alpha )+i\cos\alpha \sin(\gamma -\alpha ) \end{array}\right)$$

The polarization state of the transmission field changes with the angles of *γ* and *α*. As *γ* = 0, namely, the illuminating light is the linear polarization along the horizontal direction, the transmission field changes into (1, 0) with the rotation angle of the quarter-wave plate equaling to 0, and the transmission field is the linear polarization along the horizontal direction. Under the same illumination condition, it is exp (*i*π/2)(1, 0) as *α* = π/2, which indicates the transmission field is still the linear polarization along the horizontal direction, and as *α* = π/4 and *α* = −π/4, it changes into 2^−0.5^ exp (*i*π/4)(1, −*i*) and −2^−0.5^ exp (*i*π/4)(1, *i*), respectively, which represent the transmission fields with the right- and left-handed circular polarization states. These cases show that the quarter-wave plate can convert the horizontally polarized light into different polarization states by changing the rotation angle of the quarter-wave plate though a certain phase delay appears among the latter three cases. Certainly, the transmission polarization also changes with the value of *γ*. 

Similarly, one can obtain the transmission of one quarter-wave plate with circularly polarized light illumination, and the transmission field can be written as
(3)$$E(\alpha ,\gamma )=\frac{\sqrt{2}}{2}e^{\pm i\alpha }\left(\begin{array}{c} \cos(\alpha \mp \pi /4)\\ \sin(\alpha \mp \pi /4) \end{array}\right)$$
where the positive and negative signs correspond to the right- and left-handed circular polarization states. Obviously, the polarization direction of the transmission field changes with the value of *α*. As *α* = 0, it equals to (1, ±1), which denotes the transmission field is the linear polarization along the diagonal or anti-diagonal direction. As *α* = π/4, it changes into 2^−0.5^ exp (*i*π/4)(1, 0) or 2^−0.5^ exp (−*i*π/4)(0, 1), and as *α* = −π/4, it changes into 2^−0.5^ exp (*i*3π/4)(0, 1) or 2^−0.5^ exp (*i*π/4)(1, 0). As *α* = π/2, it changes into 2^−0.5^ exp (*i*π/2)(1, 1) and 2^−0.5^ exp (*i*π/2)(1, −1). In comparison to the case of *α* = 0, the additional phase also appears.

Therefore, when the quarter-wave plates with different rotation angles are integrated with a plane, multiple polarization states can be generated simultaneously with one certain polarization light illumination. This is just the design principle of the proposed versatile metasurface integrated polarizer. For one anisotropic nanounit, suppose the fast axis of the nanounit rotates the angle of α with respect to the *x* axis, and the transmission amplitudes along the fast and slow axis take *a_x_* and *a_y_*, the transmission of the nanounit can be expressed
(4)$$T=\left(\begin{array}{cc} a_{x}\cos^{2}\alpha +a_{y}e^{i\delta }\sin^{2}\alpha & (a_{x}-a_{y}e^{i\delta })\sin\alpha \cos\alpha \\ (a_{x}-a_{y}e^{i\delta })\;\sin\alpha \cos\alpha & a_{x}\sin^{2}\alpha +a_{y}e^{i\delta }\cos^{2}\alpha \end{array}\right)$$
where *δ* is the phase delay of nanounit along the slow axis. As *δ* = π/2 and *a_x_ = a_y_*, the nanounit can be equivalent to one quarter-wave plate. Here, we choose the L-shaped nanohole etched in the silver film deposited on the glass substrate as the nanounit to construct the metasurface, and the schematic diagram for the polarization transformation of the nanohole is shown in Figure 1a. 

We use a finite-difference time-domain method to optimize the parameters of a single L-shaped nanohole so that it can be equivalent to a quarter-wave plate. Where three parameters, namely, the arm length *L*_1_, the arm width *L*_2_, and the thickness of the silver film *H*, need to be optimized. The optimization process is achieved using the software of finite-difference time-domain (FDTD-2016) solutions. We first set the thickness of the silver film as 200 nm and sweep the arm lengths *L*_1_ and *L*_2_ from 100 nm to 500 nm with the step of 10 nm, and then we detect the amplitudes and phases of *x* and *y* components of the transmission field. While the amplitudes of *x* and *y* components are equal and the phases of *x* and *y* components have the difference of π/2, the optimization process is finished. Through the optimization, they take *L*_1_ = 280 nm, *L*_2_ = 140 nm, and *H* = 200 nm, and the L-shaped nanohole with these parameters is equivalent to a quarter-wave plate for the wavelength of 632.8 nm. 

In order to show the polarization characteristics of this nanohole, Figure 1b shows the transmission amplitudes of the optimized nanohole along the propagation direction. From the curves in Figure 1b, one can see that the amplitudes for *E_x_* and *E_y_* are equal, and their phase difference equals π/2. Obviously, the condition of *δ* = π/2 and *a_x_* = *a_y_* are satisfied, and this nanohole can be equivalent to a quarter-wave plate. The phase distributions in Figure 1c,d further confirm that the simulation results are consistent with the theoretical analysis. Under the horizontal linearly polarized light illumination, the transmission fields are just the left- and right-handed circular polarization states as the cross angle of the diagonal line of L-shaped nanohole and the *x* axis equals π/4 and −π/4. The results in Figure 1c,d also show that the fast axis of the equivalent quarter-wave plate is perpendicular to the diagonal line of the L-shaped nanohole. 

It needs to be pointed out that the optimization of parameters of L-shaped nanohole is realized with the help of the finite-difference time-domain technique. In practical simulations, the refraction index of silver is taken from the value given by Palik E D [33]. The perfect matching layers are used to avoid the reflection of boundaries. The minimum mesh takes 2 nm. The presented phase distribution is obtained at the observation plane at a distance of 2 µm away from the metasurface.

## 3. Metasurface Integrated Polarizers for Multiple Uniform Polarization States

We first design a metasurface integrated polarizer that consists of four sets of nanoholes located in four regions, and the rotation angles of each set of nanoholes take different values. Figure 2a shows the structure diagram of this metasurface integrated polarizer. The orientation angle of L-shaped nanoholes in four regions takes 0°, 45°, 90°, and 135°, respectively, and they are clearly displayed by the magnified structures, as shown in Figure 2b.

Theoretically, the designed metasurface can output four different polarization states with the horizontal linearly polarized light illumination, and the transmitted polarization states in four regions are horizontal linear polarization, right-handed circular polarization, horizontal linear polarization, and left-handed circular polarization. While the incident polarization is along the vertical direction, the polarization states in four regions are vertical linear polarization, left-handed circular polarization, horizontal linear polarization, and right-handed circular polarization. While the incident polarization is left-handed circular polarization, the transmission polarization states in four regions are linear polarization, and the polarization directions are along anti-diagonal, horizontal, diagonal directions, and vertical directions.

While the incident polarization is right-handed circular polarization, the transmission polarization states in four regions are linear polarization, and the polarization directions are along diagonal, vertical, anti-diagonal, and horizontal directions. Certainly, the transmission fields carry additional phase delays. Figure 2c gives Jones matrices of the transmission fields under four different illumination conditions, where the arrows at the left denote the illumination polarization types. Figure 2d–g shows the simulated results for the polarization distributions of this metasurface integrated polarizer. From the simulated results, one can see that the polarization distributions in four regions are almost the same as the theoretical ones. The multiple polarization states are obtained by this metasurface integrated polarizer.

At the same time, the additional phase delays in different regions of the metasurface integrated polarizer are also seen from the phase distributions for *y* components of transmission fields in Figure 2h,i where the phase difference between the red, yellow, green, and blue regions is π/4, and two regions along anti-diagonal directions are blocked because of uncertain phases, and all these results are consistent with the theoretic results shown in Figure 2c. Moreover, the intensity distributions of the *y*-component transmission fields in Figure 2j,k, where the intensity in green color parts is the largest, and the one in deep blue ones is the least, mean the intensity ratios in four regions are 1:0:1:2 for the left-handed circular polarization and 1:2:1:0 for the right-handed circular polarization. These results also exhibit the output polarization characteristics of the designed metasurface integrated polarizer.

Similarly, we design the second metasurface integrated polarizer that consists of eight sets of nanoholes located in eight regions, and the rotation angles of each set of nanoholes take different values. Figure 3a shows the structure diagram of this integrated metasurface polarizer. The orientation angles of L-shaped nanoholes in eight regions increase linearly, which are clearly shown by the magnified structures in Figure 3b. Theoretically, the polarization states in Regions 5–8 of this metasurface polarizer repeat the ones in Regions 1–4. The first and second columns in Figure 3c–f show the theoretical and simulated polarization distributions of this integrated metasurface polarizer with the illuminating light taking horizontal linear polarization, vertical linear polarization, and left- and right-handed circular polarization. One can see that the polarization states in eight regions are ascertained.

The patterns in the two right columns pattern represent the intensity distributions of *y*-component transmission fields, where the different colors denote different intensity values. For theoretical results, the intensity in white color parts is the largest, and the one in black ones is the least, and for simulation results, the intensity in green color parts is the largest, and the one in deep blue ones is the least. Comparing the theoretical and simulated results, one can see that the simulated intensity distributions are the same as the theoretical ones. The intensity distribution rules in different regions also tally with the polarization distribution rules.

## 4. Metasurface Integrated Polarizer for Nonuniform Polarization States

During the design of the above metasurface integrated polarizers, the rotation angles of nanoholes in different partitions take certain values; therefore, the polarization state at each region is uniform. Now, the rotation angles of nanoholes change with their positions, and the rotation angle of nanohole satisfies *α* = *nθ*, where *θ* denotes the position angle of the nanohole and *n* takes any integer. With the linear polarization light illumination, the transmission field of this metasurface polarizer can be rewritten as,
(5)$$E(\theta ,\gamma )=\cos n\theta \left(\begin{array}{c} \cos(\gamma -n\theta )\\ i\sin(\gamma -n\theta ) \end{array}\right)+e^{-i\frac{\pi }{2}}\sin n\theta \left(\begin{array}{c} \sin(\gamma -n\theta )\\ i\cos(\gamma -n\theta ) \end{array}\right)$$

It is obvious that the transmission field can be taken as the superposition of two elliptically polarized states with the phase difference of π/2 and the amplitudes taking cos*nθ* and sin*nθ*. This beam contains different polarization types, such as linear polarization, circular polarization, and elliptical polarization. Therefore, the polarization state is naturally nonuniform, and these kinds of polarized beams are usually called vector beams.

With the left- or right-handed circular polarization light illumination, the transmission field of this metasurface polarizer can be rewritten as,
(6)$$E(\theta )=\frac{\sqrt{2}}{2}e^{\pm in\theta }\left(\begin{array}{c} \cos(n\theta \mp \pi /4)\\ \sin(n\theta \mp \pi /4) \end{array}\right)$$

This vector beam contains the linear polarization states with different orientations, which is different from the former one, and it carries one spiral phase term with the topological charge of ±*n*. Due to the existence of the spiral phase, this vector beam is also called the vector vortex beam. In this way, we can design one highly integrated metasurface polarizer which can generate the different vector beams with different polarized light illumination.

Figure 4 shows the polarization distributions of this metasurface polarizer, with *n* taking one under horizontal linearly polarized light and left-handed circularly polarized light illumination. Under the linearly polarized light illumination, the polarization of the transmission field is linear polarization along the horizontal and vertical directions, and it is circular polarization along the diagonal and anti-diagonal directions. Under the left-handed circularly polarized light illumination, the linear polarization of the transmission field changes continuously along the azimuthal direction. These polarization characteristics are the same as the theoretical analysis.

Figure 4a,d show theoretical and simulated polarization distributions under the horizontal linearly polarized light illumination. One can see that the simulated polarization distribution is almost the same as the theoretical one, where the polarization is linear polarization along the horizontal and vertical directions, and it is circular polarization along the diagonal and anti-diagonal directions. Figure 4b,e show theoretical and simulated intensity distributions for *y*-component transmission fields, and they tally with the rule of sin^2^(2*θ*). The theoretical and simulated step phase distributions shown in Figure 4c,f indicate the characteristic of a pure vector beam.

Under the circularly polarized light illumination, this highly integrated metasurface polarizer can generate the vector vortex beam, and the theoretical and simulated polarization distributions shown in Figure 4g,j give the sound verification. Where the illumination polarization takes the left-handed circular polarization. One can easily see that the orientation of the linear transmission polarization continuously changes with the azimuthal angle. The theoretical and simulated *y*-component intensity distributions tally with the rule of sin^2^(*θ* − π/4). Moreover, the theoretical and simulated phase distributions shown in Figure 4i,l show the phase increases 2π along the clockwise direction. It indicates the vortex the topological charge equal to one generates. These results verify the generation of vector vortex beam.

## 5. Conclusions

In conclusion, we propose two kinds of versatile integrated polarizers based on optical metasurfaces consisting of L-shaped nanoholes. One can generate the combination of uniform polarization states distributing different partitions, and the combination ways of uniform polarization states can be adjusted by changing the orientation angle of nanoholes in different partitions. The other can generate the vector beam or vector vortex beam. The generated polarization types, including linear polarization, circular polarization, and elliptic polarization, change with the azimuthal angle. Moreover, the same integrated polarizer may output a different polarization combination or a different vector beam, which depends on the illumination condition. A multiple polarization output, compact and simple structure, and feasible operation of metasurface integrated polarizer are beneficial to more applications of the polarization optics in optical imaging, biomedical sensing, and other fields.

## Figures and Tables

**Figure 1 nanomaterials-12-02816-f001:**
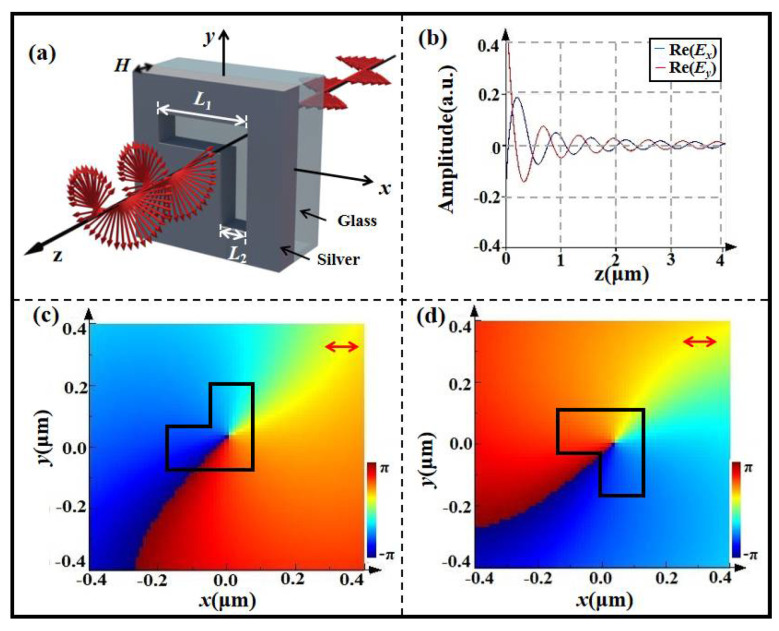
(**a**) Schematic diagram for polarization transformation of nanohole, (**b**) the transmission amplitude and (**c**,**d**) phase distributions of nanohole with its orientation directions along diagonal and anti-diagonal directions.

**Figure 2 nanomaterials-12-02816-f002:**
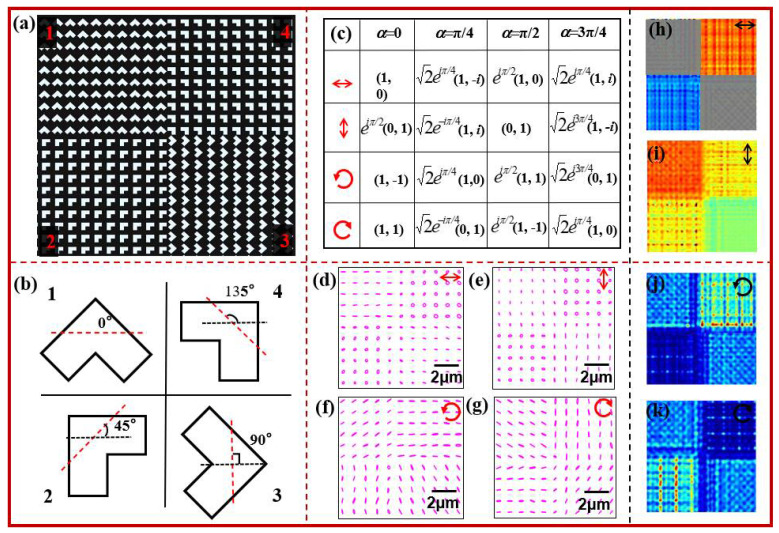
(**a**,**b**) Structures of metasurface integrated polarizer with four partitions where L-shaped nanoholes have the different orientation angles, (**c**) the transmission fields of nanohole with different orientation angle under different polarization light illumination, (**d**–**g**) the simulated polarization distributions of metasurface integrated polarizer, (**h**,**i**) are the simulated phase distributions for *y* components of transmission fields and (**j**,**k**) the simulated intensity distributions for *y* components of transmission fields, where the inserted arrows denote the illumination polarization states, and the scale bar denotes 2 µm.

**Figure 3 nanomaterials-12-02816-f003:**
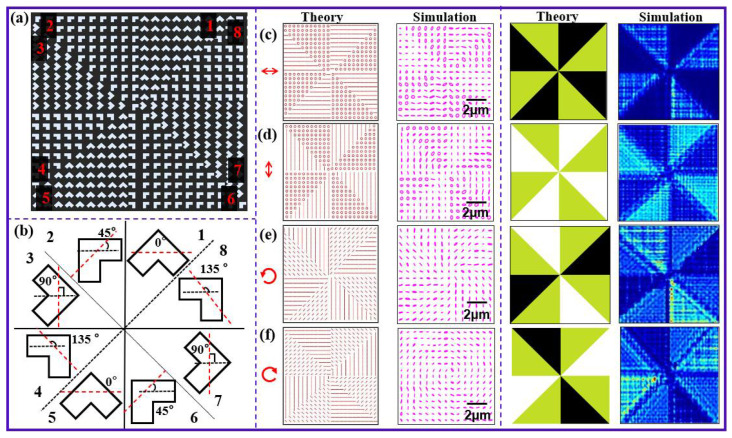
(**a**,**b**) Structures of metasurface integrated polarizer with four partitions, and (**c**–**f**) theoretical (at the first column) and simulated (at the second column) polarization distributions of the metasurface integrated polarizer. The patterns on two right columns represents the theoretical and simulated values of intensity in the *y* direction. The scale bar denotes 2 µm.

**Figure 4 nanomaterials-12-02816-f004:**
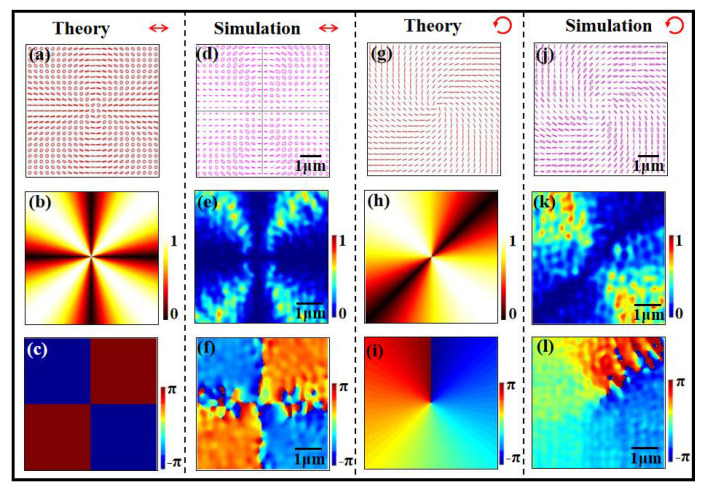
Theoretical and simulated polarization distributions (**a**,**d**,**g**,**j**), *y*-component intensity (**b**,**e**,**h**,**k**) and phase distributions (**c**,**f**,**i**,**l**) of highly integrated metasurface polarizer the under horizontal linearly polarized light and left-handed circularly polarized light illumination. The scale bar denotes 1 µm.

## Data Availability

Not applicable.

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
