# Peer review of "Versatile Integrated Polarizers Based on Geometric Metasurfaces"

_nanomaterials, 2022, doi:10.3390/nano12162816_

Round 1
Reviewer 1 Report
Line 24: "applications and flexible" should be "applications with flexible"
LIne 33: "outputting" should read "outputs"
Line 34: replace "and the output of controllable polarization states" with " that"
Line 48: Replace "few" with "little"
Line 71: Equations number (2) should be right adjusted
Author Response
Comment: The reviewer suggests that Line 24: "applications and flexible" should be "applications with flexible", Line 33: "outputting" should read "outputs", Line 34: replace "and the output of controllable polarization states" with " that", Line 48: Replace "few" with "little", and Line 71: Equations number (2) should be right adjusted.
Reply: On line 24, there are two sentence between applications and flexible. For clearness, we depart them by one comma. For the sentence on lines 33 and 34, we rewrite it as “the multifunctional polarizer that outputs simultaneously multiple polarization states is fascinating for the practical applications”. The word of few on line 47 and the number of equation on line 70 have modified in the revised manuscript. All the modifications are written in red.
Reviewer 2 Report
The authors present a theoretical study describing the design and evaluation of a metamaterial-based polarizers.
The study is well argued and the presented data supports the authors conclusion. Moderate English editing would be beneficial, since there are typos and grammar errors that make following the arguments difficult.
I recommend the following addition/changes before publishing of the manuscript:
This is a theoretical study. Analytic concepts are used to design metasurface structures. Optical properties are inferred from analytic equations and the structures are then simulated using FDTD. The authors need to share more details about the FDTD calculations, as these simulations are highly dependent on the simulation parameters. The current manuscript only discusses the method vaguely.
For example, how have the L-shaped patterns been optimized? Which code was used?
I recommend a separate section/paragraph describing the FDTD set-up and data interpretation in more detail.
The authors should avoid language such as “verified”. This is a theoretical study, so I think the correct language would be that “simulation agrees with theory”. There are no experimental verifications of any of the presented data.
Figure 2 and 3 need improvement. The inserts in the subpanels 2 (d-g) and 3 (c-f) are too small to interpret. I understand from the discussion in the manuscript that the data they depict is very important and aides the discussion, thus I recommend a separate panel for each of these datasets.
The final structure, with continuously varying angles of the L-shaped metasurface unit needs to be described better. I can follow the description in the text, however this is needlessly difficult for the reader. Figure 4 only discusses results, not the geometry per se. The authors have done an excellent job in visualizing their designs in figures 2 and 3. A similar depiction of the final structure would elevate the presentation in this paper.
Author Response
According to the comments given by Reviewer 2, we give the following modifications.
Comment 1: The reviewer points out that “This is a theoretical study...... The current manuscript only discusses the method vaguely. For example, how have the L-shaped patterns been optimized? Which code was used? I recommend a separate section/paragraph describing the FDTD set-up and data interpretation in more detail.”
Reply: The L-shaped nanohole etched on the silver film needs to be equivalent to a quarter wave plate so as to effectively control the polarization state of light field. The parameters of L-shaped nanohole should be optimized to satisfy this condition. The optimization process is to find the exact size of L-shaped hole and the thickness of silver film in terms of the numerical simulation. The simulation calculation is achieved using the business software of Finite-difference time-domain difference solutions. Thus, we sweep these parameters of L-shaped nanohole and detect the amplitudes and phases of x and y components of the transmitted field. When the amplitudes of x and y components are equal and their phase difference equals to π/2, the parameters of L-shaped nanohole are ascertained. The optimization process of L-shaped nanohole is described in detail from line 107 to line 115 of Section 2 of the revised manuscript.
Comment 2: The reviewer suggests that “avoid language such as “verified”. This is a theoretical study, so I think the correct language would be that “simulation agrees with theory”. There are no experimental verifications of any of the presented data.”
Reply: We modify the word “verifies” on line 57, “verified" on line 128 and verify on line 184 in the revised manuscript.
Comment 3: The reviewer suggests that “Figure 2 and 3 need improvement. The inserts in the subpanels 2 (d-g) and 3 (c-f) are too small to interpret. I understand from the discussion in the manuscript that the data they depict is very important and aides the discussion, thus I recommend a separate panel for each of these datasets.”
Reply: For the inserted patterns in Figure 2, we list them as Figs. 2h-2k in revised manuscript. The corresponding explanations in the paragraph before Fig.3 and in the caption of Fig.2 are also modified. For the inserted patterns in Figure 3, we list them as two lines at the right of Figs. 3c-3f in revised manuscript. The corresponding explanations in two paragraphs after Fig.3 and in the caption of Fig.3 are also modified.
Comment 4: The reviewer points out that “The final structure, with continuously varying angles of the L-shaped metasurface unit needs to be described better. I can follow the description in the text, however this is needlessly difficult for the reader.”
Reply: While the nanoholes rotate with their positions, the description is simplified on lines 213 and 214 in revised manuscript.
All the modification are written in red words.